# Specific GPCRs elicit unique extracellular vesicle miRNA array signatures

Xiao Shi[1]*, Michelle C Palumbo[2], Sheila Benware[1], Jack Wiedrick[3], Sheila Markwardt[3], Aaron J Janowsky[1,2,4]

[1]Research Service, Veterans Affairs Portland Health Care System, Portland, United States; [2]Department of Behavioral Neuroscience, Oregon Health and Science University, Portland, United States; [3]Biostatistics and Design Program, OHSU-PSU School of Public Health, Oregon Health and Science University, Portland, United States; [4]Department of Psychiatry, Oregon Health and Science University, Portland, United States

## eLife Assessment

This study presents **valuable** findings by demonstrating that specific GPCR subtypes induce distinct extracellular vesicle miRNA signatures, highlighting a potential novel mechanism for intercellular communication with implications for receptor pharmacology within the field. The evidence is **solid**, however, more experiments are needed to determine whether the distinct extracellular vesicle miRNA signatures result from GPCR-dependent miRNA expression or GPCR-dependent incorporation of miRNAs into extracellular vesicles.

**Abstract** All cells secrete extracellular vesicles (EVs) containing nucleic acid cargo, including microRNAs (miRNAs), that regulate the function of receiving cells. G protein-coupled receptors (GPCRs) affect intracellular function via multiple signaling cascades. However, the mechanisms of GPCR intercellular signaling through EV miRNA activity remain unknown. Human U2 osteosarcoma cells expressing native GPCRs were used to selectively stimulate distinct G protein signaling cascades ($G\alpha_i$, $G\alpha_q$, $G\alpha_{12/13}$, and β-arrestin) by members of specific receptor subclasses, including the adenosine receptor A1 (ADORA1), the histamine receptor H1 (HRH1), the frizzled class receptor 4 (FZD4), and the atypical chemokine receptor 3 (ACKR3), respectively. We hypothesized that stimulation of specific classes of GPCRs would cause the release of EVs containing miRNAs with receptor-specific up- or downregulated expression, affecting unique pathological downstream signaling cascades. Receptor-specific agonists dose-dependently increased respective signaling cascade intermediates. We found no change in the quantity of EVs (~200 nm diameter), but there were distinct EV miRNA signatures following stimulation of GPCRs. Network analyses of differentially expressed miRNA and their predicted targets validated the linkage between specific receptors and cell function and pathological states. The data can be used to reverse engineer mechanisms involving EV miRNAs for various physiological and pathological processes. GPCRs are major pharmacological targets, so understanding the mechanisms that stimulate or inhibit GPCR-mediated changes in extracellular miRNA signatures could improve long- and short-term therapeutic and unwanted drug effects.

## Introduction

G protein-coupled receptors (GPCRs) comprise a protein superfamily with five major branches: glutamate, rhodopsin, adhesion, frizzled/taste 2, and secretin (*Stevens et al., 2013*). These transmembrane

*For correspondence: shixiaosmile@yahoo.com

Competing interest: The authors declare that no competing interests exist.

proteins transduce neurotransmitter signals from outside the cell to affect internal cellular function (*Stevens et al., 2013*). Translation of extracellular signals to intracellular biochemical cascades is a multistep process involving specific receptor-ligand interactions, conformational changes allowing interactions with regulatory G proteins, such as $G\alpha_i$, $G\alpha_q$, $G\alpha_{12/13}$, and β-arrestin, and activation of specific effector systems (*Asher et al., 2022*; *Heydenreich et al., 2023*). GPCR function or malfunction is implicated in numerous human diseases, and approximately 700 medications are designed to target GPCRs and related proteins (*Sriram and Insel, 2018*) and to alter cellular activity. In addition to the canonical intracellular protein cascades, GPCR signaling is both a precursor and effector of extracellular mechanisms of communication (*Bebelman et al., 2020*).

Cells interact through the secretion and uptake of extracellular vesicles (EVs) (*Rothman, 1996*). EVs are a heterogeneous population of membrane vesicles that are classified into exosomes, microvesicles, and apoptotic bodies based on their biogenesis, release pathways, size, content, and function (*Doyle and Wang, 2019*; *Palmisano et al., 2012*). EVs transport a range of molecules such as nucleic acids, including microRNAs (miRNA), lipids, and proteins associated with the cell lipid membrane such as GPCRs (*Kahn et al., 2017*; *Mack et al., 2000*). GPCRs and EVs have interacting functions since EV biogenesis and release can be regulated by GPCR stimulation, and EVs can contribute to GPCR signaling (*Kriebel et al., 2018*; *Verweij et al., 2018*). In fact, a number of GPCRs alter EV release and/or EV cargo (*Bhattacharya et al., 2023*; *Conrad et al., 2020*; *Lee et al., 2023*). However, the effects of stimulating specific GPCR subclasses, in head-to-head comparisons, on EV content and subsequent alteration of receiving cell function remain to be elucidated (*Tetta et al., 2013*).

miRNAs are small non-coding RNAs (*Tabara et al., 2002*) and play crucial roles in plant and animal physiology, including development (*Zhang et al., 2015*). They are implicated in symptoms of pathology at the intersection of substance use and HIV (*Odegaard et al., 2020*), Alzheimer's disease (*Sandau et al., 2022*), and diabetes (*Chao et al., 2023*). miRNAs function by translationally repressing target mRNA transcripts. The human genome includes approximately 2600 miRNAs, and individual miRNAs can regulate tens to hundreds of gene targets (*Plotnikova et al., 2019*). Therefore, miRNAs are one of the largest classes of regulatory RNAs that affect complex genetic networks. However, whether GPCRs are post-transcriptionally affected by miRNAs is controversial (*Ofer and Linial, 2022*; *Orr-Burks et al., 2021*).

Many drugs that stimulate GPCRs affect EV miRNA disposition. For example, ethanol potentiates GABA-A receptor function and alters EV miRNA expression (*Bourgeois et al., 2023*). Methamphetamine use alters groups of human plasma EV miRNAs (*Sandau et al., 2020*). Bombesin receptor subtype 3 stimulation results in its own incorporation into EVs and alteration of RhoA function in receiving cells (*Wang et al., 2022b*). GPCRs affect EV secretion and miRNA expression pertaining to platelet aggregation, function, and activation (*Ambrose et al., 2018*; *Jin et al., 2022*). Additionally, stimulation of CXCR7, a chemokine receptor, results in expression changes of specific miRNAs related to colorectal cancer metastasis. However, it is unknown if stimulation of specific GPCR subclasses affects EV miRNA content to alter downstream cellular communication, as well as signal cascades involved in physiology and disease. Using U2OS cells, we examined the effects of stimulating the adenosine receptor A1 (ADORA1), the histamine receptor H1 (HRH1), and atypical chemokine receptor 3 (ACKR3), members of the rhodopsin-like GPCR family, as well as the frizzled class receptor 4 (FZD4). These receptors are $G\alpha_i$-, $G\alpha_q$-, β-arrestin-, and $G\alpha_{12/13}$-coupled, respectively. Stimulation resulted in the release of EVs with unique miRNA signatures. Network analyses of EV miRNA target transcripts showed both up- and downregulation of protein cascades involved in physiological and pathological states of cell function. This work demonstrates the large-scale genetic influence of miRNA arrays as a product of GPCR stimulation and can be used to characterize downstream pharmacological effects of receptor-drug interactions.

## Results

### GPCR expression in U2OS cells

To characterize the effect of GPCR stimulation on EV miRNA content, we examined GPCR expression in different cell lines from The Human Protein Atlas (*Karlsson et al., 2021*). We determined that the human osteosarcoma cell line, U2OS, likely expressed receptors from multiple branches of the GPCR family tree (https://www.proteinatlas.org/search/U2OS+cells), including rhodopsin, adhesion, and

frizzle/taste receptors, each associated with differing G-protein cascade activation (*Supplementary file 1*). Numerous other cell lines, including U-251MG, A-431, and SH-SY5Y, were considered but not used because they did not highly express a significant spread of receptor subtypes (*Supplementary file 1*). Additionally, we avoided the use of transfected or overexpressing cells because the transfection and selection protocols could have their own effects on EV synthesis, expression, and cargo selection (*Im et al., 2019*; *Németh et al., 2017*; *Nordin, 2022*). The protein expression data of U2OS cells provided by the Human Protein Atlas database indicated that ADORA1, HRH1, ACKR3, and FZD4 were expressed at levels sufficient to demonstrate second messenger system activity, i.e., function.

The expression of these receptors in U2OS cells was further validated through various functional assays. 2-Chloro-$N^6$-cyclopentyladenosine (CCPA), an ADORA1 selective agonist, dose-dependently inhibited forskolin-induced cAMP levels (*Figure 1A*). The inhibition of cAMP by CCPA was reduced by antagonist 8-cyclopentyl-1,3-dipropylxanthine (DPCPX), suggesting that the decreased cAMP level resulted from ADORA1 activation. The activation of HRH1 was assessed by measuring inositol monophosphate (IP1) accumulation following 2-pyridylethylamine dihydrochloride (PEA) stimulation, and the effect was abolished by the HRH1 antagonist cetirizine (*Figure 1B*). To evaluate the function of FZD4, we measured the activity of alkaline phosphatase (ALP), the downstream gene product of the Wnt/β-catenin signaling cascade. Norrin, another FZD4 agonist, has high affinity (*Xu et al., 2004*). However, Norrin is difficult to purify from mammalian sources (*Perez-Vilar and Hill, 1997*). We found the commercially available Norrin (R&D Systems) did not stimulate FZD4 (data not shown). Although Wnt family member 3A (Wnt3A) can bind to several members of the frizzled receptor family, it has much higher affinity binding at FZD4 than FZD8 (*Kozielewicz et al., 2021*), and both are expressed in U2OS cells. Therefore, we used Wnt3A as the FZD4 ligand. Wnt3A significantly increased ALP activity compared to the vehicle control (*Figure 1C*). The endogenous chemokine ligand stromal-derived factor-1α (SDF-1α) induced ERK1/2 phosphorylation through chemokine receptor, ACKR3 (*Figure 1D*). Phospho-ERK1/2 levels were normalized to total ERK1/2 as shown on the immunoblot (*Figure 1—figure supplement 1*). Thus, we selected a dose of 10 μM of CCPA, 200 μM of PEA, 100 ng/ml of Wnt3A and SDF-1α to selectively stimulate ADORA1, HRH1, FZD4, and ACKR3, respectively.

## EV characterization from U2OS cells

First, we evaluated native U2OS EVs isolated from the culture media using ultrafiltration followed by size exclusion chromatography (SEC). The immunoblot revealed the presence of EV surface markers, tetraspanins CD9, CD63, and CD81 (*Figure 2A*), in SEC fractions 7, 8, and 9. We also detected EV cytosolic markers syntenin (*Kugeratski et al., 2021*) and flotillin-1 in individual fractions 7–9. In addition, calnexin, an endoplasmic reticulum marker and a negative EV control, was absent in fractions 7–9 (*Figure 2A*). All EV markers were absent in the media control sample, which suggested that the SEC fractions 7–9 were enriched in EVs from U2OS cells. The pooled EV-enriched fractions 7–9 were assessed for vesicle shape and size by transmission electron microscopy (TEM), which revealed vesicular structures ranging from 100 to 200 nm (*Figure 2B*, red arrows). We quantified the EV concentration and size distributions using fluorescence nanoparticle tracking analysis (f-NTA), which enhances the detection of membrane vesicles by incorporation of a fluorophore into the EV lipid membrane. The average concentration of EVs in U2OS culture supernatant was $6 \times 10^9$ particles/ml (*Figure 2C and D*), which was significantly increased compared to the media control. Despite using EV-depleted fetal bovine serum (FBS), some residual bovine EVs, especially small-size EVs (*Pham et al., 2021*), remained in the media control, with an average concentration of $4.45 \times 10^9$ particles/ml (*Figure 2C and D*). However, they did not exhibit the EV markers we examined (*Figure 2A*). Thus, those vesicles are most likely not derived from the host cell (*Lehrich et al., 2021*). Therefore, media control samples were included in all subsequent experiments to avoid any miRNA artifacts caused by serum. The size of EVs ranged from 50 to 500 nm with no significant difference in the median diameter between the media control (214±14.41 nm) and U2OS-exposed media (237±5.17 nm) (*Figure 2C*). We also compared the EV concentration in various GPCR agonist treatment groups with their vehicle controls. Notably, the EVs from agonist-treated samples and vehicle controls exhibited similar concentration and size, with median diameter ranging from 210 to 240 nm (*Figure 2—figure supplement 1*). Therefore, the pooled EV-enriched SEC fractions 7–9 were used for further miRNA assessment.

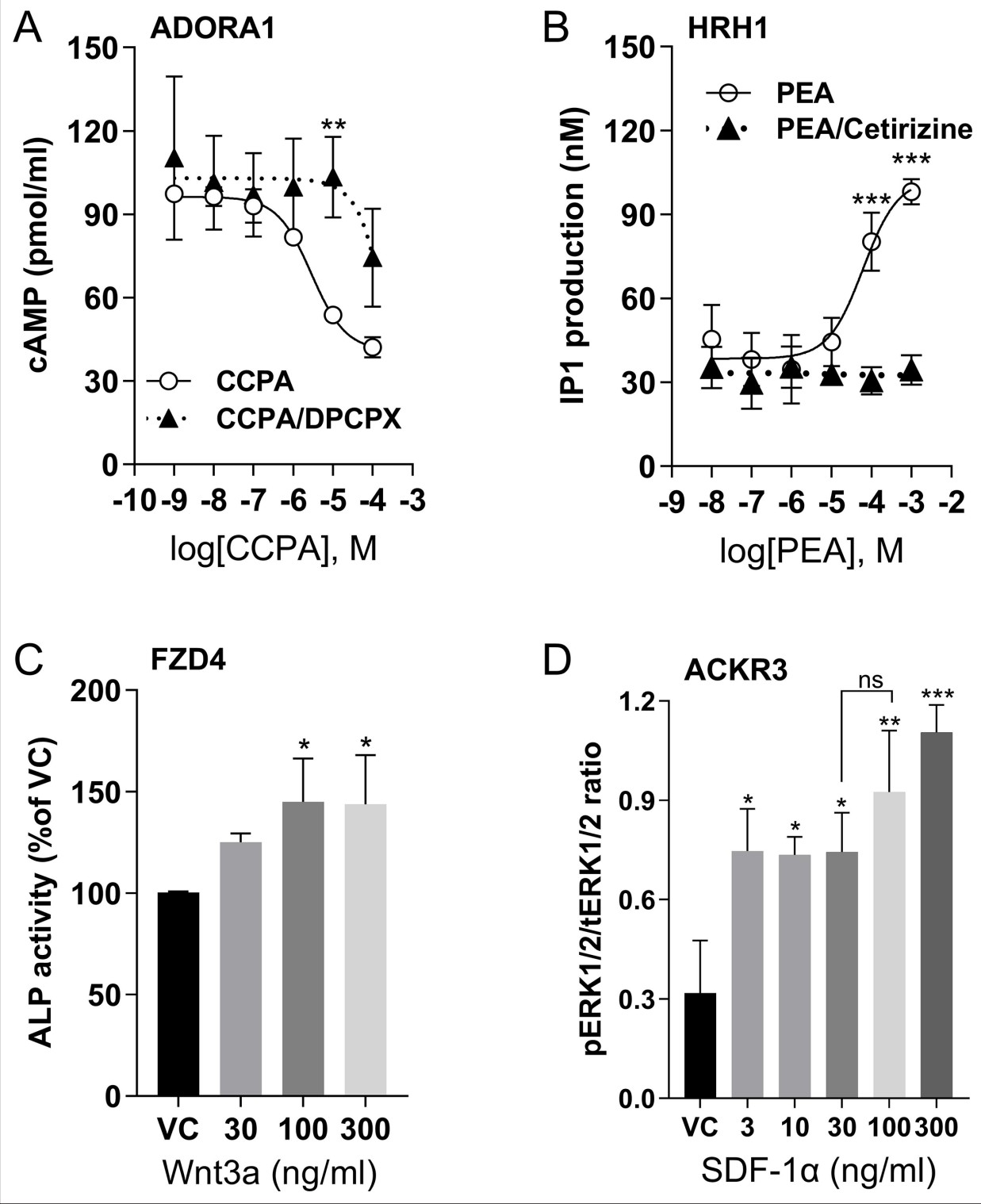

**Figure 1.** G protein-coupled receptor (GPCR) activation in U2OS cells by selective agonists. (**A**) Dose-dependent cAMP accumulation after ADORA1 activation with agonist, 2-chloro-$N^6$-cyclopentyladenosine (CCPA), or inhibition with antagonist 8-cyclopentyl-1,3-dipropylxanthine (DPCPX) (5 µM). (**B**) Dose-dependent IP1 accumulation after 2-pyridylethylamine dihydrochloride (PEA) stimulation of histamine receptor H1 (HRH1) and inhibition by cetirizine (1 µM). (**C**) Alkaline phosphatase (ALP) activity after stimulation of frizzled class receptor 4 (FZD4) by Wnt3a. (**D**) The ratio of phosphorylated ERK1/2 (pERK1/2) by total ERK 1/2 (tERK1/2) after stimulation of ACKR3 by SDF-1α. For all graphs, data are shown as mean ± SD, with n=3 independent repeats, each having duplicate determinants, ns = not significant, *p<0.05, **p<0.01, ***p<0.001 vs. vehicle control (VC). Statistical significances were determined by one-way or two-way ANOVA of receptors by agonists or antagonists and post hoc Tukey's or Sidak's testing for multiple comparisons.

*Figure 1 continued on next page*

*Figure 1 continued*

The online version of this article includes the following source data and figure supplement(s) for figure 1:

**Figure supplement 1.** Representative immunoblots show stimulation of ACKR3 with SDF-1α-induced phosphorylation of ERK1/2 (pERK1/2, 44kDa/42kDa) compared to total ERK1/2 (tERK1/2, 44kDa/42kDa).

**Figure supplement 1—source data 1.** Original western blots for *Figure 1—figure supplement 1*.

**Figure supplement 1—source data 2.** PDF file containing the original western blots for *Figure 1—figure supplement 1*, with relevant bands and experimental conditions labeled.

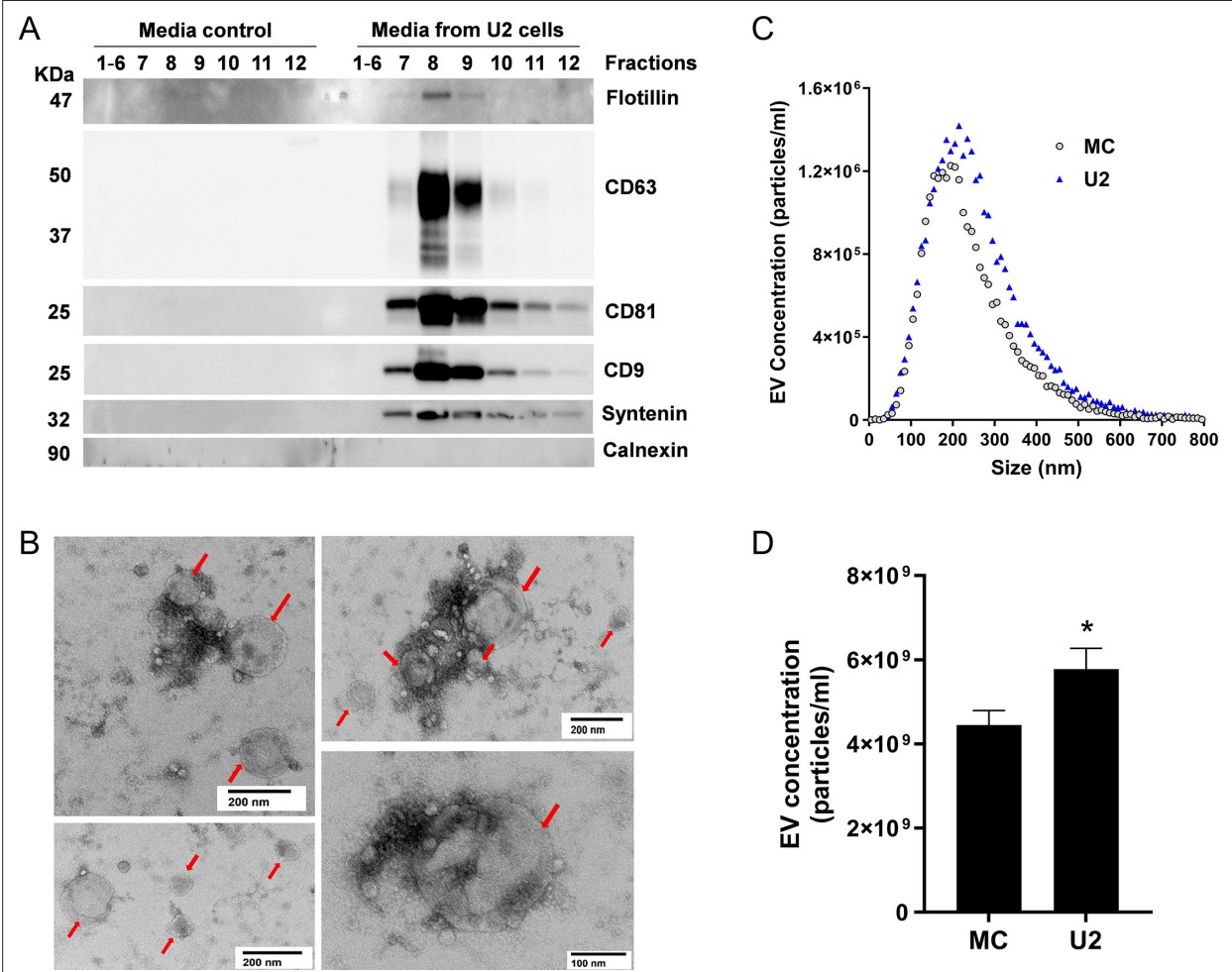

**Figure 2.** Characterization of extracellular vesicles (EVs) isolated from U2OS cell culture media. (**A**) Representative immunoblots of EVs isolated by size exclusion chromatography (SEC) from U2OS cell culture media (U2) but not from media control (MC) show detection of EV markers, including CD9, CD63, CD81, flotillin, and syntenin, in EV fractions 7–9. Endoplasmic reticulum marker, calnexin, was not detected in the EV fractions. (**B**) Representative images from transmission electron microscopy of isolated EVs (red arrows) from pooled EV-enriched fractions 7–9. (**C**) Size distribution of pooled EV fractions 7–9 isolated from MC and U2. (**D**) Quantification of pooled EV fractions 7–9 of MC and U2 measured by fluorescence nanoparticle tracking analysis (f-NTA). Data are shown as mean ± SEM, n=7–8. A Student's t-test determined significant differences in the EV concentration between the MC and U2, *p<0.05.

The online version of this article includes the following source data and figure supplement(s) for figure 2:

**Source data 1.** Original western blots for *Figure 2A*.

**Source data 2.** PDF file containing the original western blots for *Figure 2A*, with relevant bands and experimental conditions labeled.

**Figure supplement 1.** Quantification of extracellular vesicles (EVs) in U2OS media in response to receptor activation.

## EV miRNAs differentially expressed following GPCR stimulation

We utilized high-throughput miRNA arrays to identify EV miRNA signatures following receptor activation. To analyze the EV miRNAs from stimulated receptors and avoid any confusion posed by the miRNA content in the media from non-stimulated cells, we excluded any miRNAs that were only present in the media control (*Supplementary file 2*) or were changed less than 10-fold (<3.3 PCR cycles) between the media control and vehicle control. This allowed us to focus on analyzing specific receptor stimulation effects. We identified 105 miRNAs associated with ADORA1, 92 with HRH1, 88 with FZD4, and 86 with ACKR3 that expressed in at least 80% of either vehicle control or the agonist-treated samples and underwent statistical analysis. To further narrow down our miRNA of interest after receptor stimulation, we performed differential expression testing using within-pairs linear regression and calculated p-values using a nonparametric rank-based (Skillings-Mack) test. This approach provided unbiased estimates of stimulation treatment effects. We subsequently ranked the miRNAs within each receptor type by magnitude of effect and selected all miRNAs with Skillings-Mack p-value<0.2 for further investigation, emphasizing discovery power over avoidance of false positive results. Any miRNAs with absolute fold change ≥1.5 ($|\Delta\Delta Cq|$≥0.585) were chosen for enrichment analyses.

We directly compared the miRNA expression across each receptor. The heatmap illustrated distinct EV miRNA patterns unique to each receptor (*Figure 3A*). Our analysis showed that 29 miRNAs were differentially expressed in the ADORA1 group compared to its vehicle control, with 10 miRNAs demonstrating an absolute fold change greater than 1.5 (*Figure 3B*). Furthermore, 5 out of 10 miRNAs were observed with p<0.05 (*Table 1*). For the HRH1 group, 3 out of 4 differentially expressed miRNAs were found to have p<0.05 (*Figure 3C* and *Table 1*). In contrast, among the 22 differentially expressed miRNAs in the FZD4 group, 10 exhibited an absolute fold change greater than 1.5 (*Figure 3D*), but only one miRNA (miR-203a-3p) met the threshold of p<0.05 (*Table 1*). A similar pattern was also observed in the ACKR3 group (*Figure 3E* and *Table 1*). Additionally, we further assessed the shared and non-shared changes in miRNAs with at least 1.5-fold change in either direction for each receptor (*Figure 3F*). We found that miR-550a-5p, miR-502-3p, miR-137, and miR-422a were the most changed miRNAs following ADORA1, HRH1, FZD4, and ACKR3 stimulation, respectively (refer to *Supplementary file 3* for a complete list of all differentially expressed miRNAs of individual receptors and raw data in data availability section).

## miRNA targets and pathway predictions following GPCR stimulation

We next performed target prediction and functional annotations on all miRNAs of interest that exhibited an absolute fold change ≥1.5 (*Figure 3*) for each receptor. For analysis involving multiple miRNAs, we selected gene targets predicted by at least two miRNAs for functional analysis through the Kyoto Encyclopedia of Genes and Genomes (KEGG) database. Following ADORA1 stimulation, we identified 3562 miRNA predicted target genes, with 709 predicted genes targeted by at least two differentially regulated miRNAs. The top 25 enriched KEGG pathways associated with ADORA1 miRNA targets are primarily related to cell proliferation, apoptosis, and cancer (*Figure 4A*). After HRH1 stimulation, 357 predicted miRNA targets were selected for enrichment analysis. Many of the top predicted miRNA target enriched KEGG pathways were linked to various cancers (*Figure 4B*). We observed significant enrichment of FoxO and ErbB signaling (*Figure 4B*), which play crucial roles in angiogenesis, cell proliferation, apoptosis, and modulation of the immune response (*Appert-Collin et al., 2015*; *Kim et al., 2022*; *Lam et al., 2013*; *Ryzhov et al., 2017*) and are associated with HRH1 function (*Fernández-Nogueira et al., 2018*; *Hatipoglu et al., 2023*; *Li et al., 2022*). Additionally, we found that HRH1 miRNA predicted targets are enriched in endocrine and nervous system pathways. For instance, several hub genes identified in HRH1 miRNA target PPI network, such as PTEN, MYC, GSK3B, and SMAD3 (*Figure 4—figure supplement 1B*), are known to be involved in insulin signaling (*Cheung et al., 2010*; *Li et al., 2020*; *Patel et al., 2008*; *Wang et al., 2022a*). The most upregulated miRNA from HRH1 stimulation, miR-502-3p, was associated with multiple enriched pathways related to insulin resistance, including EGFR tyrosine kinase inhibitor resistance, AGE-RAGE signaling pathway in diabetic complications, and pancreatic cancer (*Figure 4—figure supplement 2A*). Overall, these findings suggest that altered miRNA expression following HRH1 activation may play a role in diabetes.

Generally, the miRNA targets of FZD4 were related to cancer, drug resistance, embryonic development, and aging (*Figure 4C*). The most downregulated miRNA after FZD4 stimulation, miR-137,

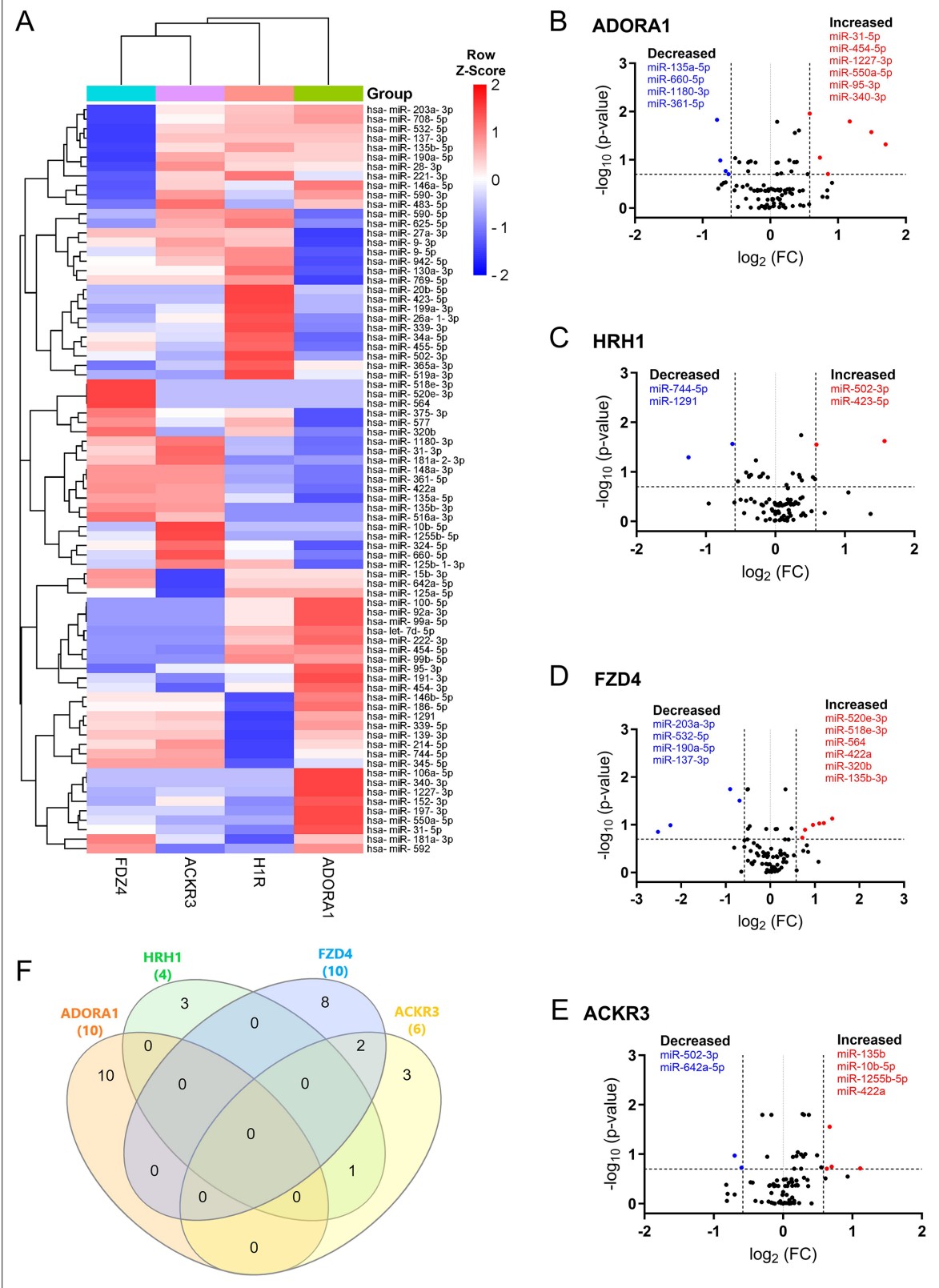

**Figure 3.** Differentially expressed extracellular vesicle (EV) microRNAs (miRNAs) in response to G protein-coupled receptor (GPCR) activation. (**A**) The heatmap shows unsupervised hierarchical clustering of miRNA expression (row) following GPCR stimulation (column). The relative abundance of miRNAs is represented in Z-score value (z-transformed fold changes); blue, below-mean expression; red, above-mean expression. (**B–E**) Volcano plots display the analysis of EV miRNAs after ADORA1 (**B**), HRH1 (**C**), FZD4 (**D**), and ACKR3 (**E**) activation. On the x-axis, the dotted line indicates miRNAs that satisfied

*Figure 3 continued*

the |log$_2$ fold change|≥ 1.5 cutoff, while the dotted line on the y-axis indicates miRNA that met Skillings-Mack p-value<0.2. (**F**) Venn diagram shows the miRNAs of interest with at least 1.5-fold change across four GPCR groups following treatment. n=5–6 replicates per GPCR group.

The online version of this article includes the following figure supplement(s) for figure 3:

**Figure supplement 1.** Sensitivity analysis across alternative estimates of treatment effects.

had targets that are enriched in GABAergic and dopaminergic synapses, suggesting its role in the regulation of synaptic function (***Figure 4—figure supplement 2B***). Additionally, miR-137 targets were linked to substance abuse and neurodegenerative diseases (***Figure 4—figure supplement 2B***). MiR-135b-3p and miR-422a exhibited increased expression following FZD4 and ACKR3 activation (***Figure 3D and E*** and ***Supplementary file 3***). Therefore, some of the enriched pathways, including various cancer types and signaling pathways, shared similarities between two receptors (***Figure 4C and D***). Alongside cancer-related pathways, several enriched pathways related to aging and neuro-degenerative diseases, such as cellular senescence and autophagy, were identified as ACKR3 miRNA targets (***Figure 4D***). Insulin resistance-related pathways were also observed when analyzing miRNA targets following ACKR3 stimulation (***Figure 4D***). Note, however, that the top identified hub genes between the two receptors were distinct (***Figure 4—figure supplement 1***), suggesting FZD4 and ACKR3 may regulate the same cellular process through different mechanisms.

## Discussion

We hypothesized that stimulation of GPCRs can regulate the EV miRNA profile. We used a combined approach of ultrafiltration and SEC to isolate EVs from agonist-stimulated culture media, then followed up with miRNA qPCR arrays. Our findings provide evidence that within a single cell type, head-to-head comparisons of specific agonist stimulation of GPCR subclasses reveal unique EV miRNA profiles.

Although various methods exist for isolating EVs, obtaining a high yield of pure EVs with minimal contamination is challenging (***Sidhom et al., 2020***). Combining ultrafiltration with SEC allowed us to isolate EVs with sufficient yield and purity that are suitable for further EV miRNA characterization. We noticed the median size of EVs measured by nanoparticle tracking analysis (NTA) was larger than the size reported by TEM. While f-NTA was utilized in this study to enhance membrane vesicle detection, it may not differentiate EVs from other similar particles, such as protein aggregates and lipoproteins. This overlap could lead to overestimating the EV size and impacting the accuracy of EV detection. Furthermore, this discrepancy could be due to EV storage and freeze-thaw cycles (***Gelibter et al., 2022***; ***Görgens et al., 2022***). In contrast to freshly prepared samples for TEM, EV aggregation after the freeze-thaw cycle could result in increased particle size detected by NTA. Contrary to other studies that have reported increased EV release after GPCR stimulation (***Ambrose et al., 2018***; ***Conrad et al.,***

**Table 1.** Extracellular vesicle (EV) microRNAs (miRNAs) differentially expressed in four G protein-coupled receptor (GPCR) groups. Differentially expressed miRNAs (p<0.05) were identified in each treatment group compared to its vehicle control, n=5–6 samples of each group.

| GPCRs | miRNAs | log$_2$ (FC) | p-Value | SD of log$_2$(FC) |
|---|---|---|---|---|
| | hsa-miR-550a-5p | 1.70 | 0.0455 | 0.22 |
| | hsa-miR-1227-3p | 1.49 | 0.0253 | 0.16 |
| | hsa-miR-454-5p | 1.17 | 0.0143 | 0.27 |
| | hsa-miR-31-5p | 0.58 | 0.0143 | 0.07 |
| ADORA1 | hsa-miR-135a-5p | −0.79 | 0.0143 | 0.18 |
| | hsa-miR-502-3p | 1.57 | 0.025 | 0.26 |
| | hsa-miR-423-5p | 0.59 | 0.025 | 0.16 |
| HRH1 | hsa-miR-744-5p | −0.62 | 0.025 | 0.14 |
| FZD4 | hsa-miR-203a-3p | −0.90 | 0.0143 | 0.16 |
| ACKR3 | hsa-miR-135b-3p | 0.67 | 0.0253 | 0.11 |

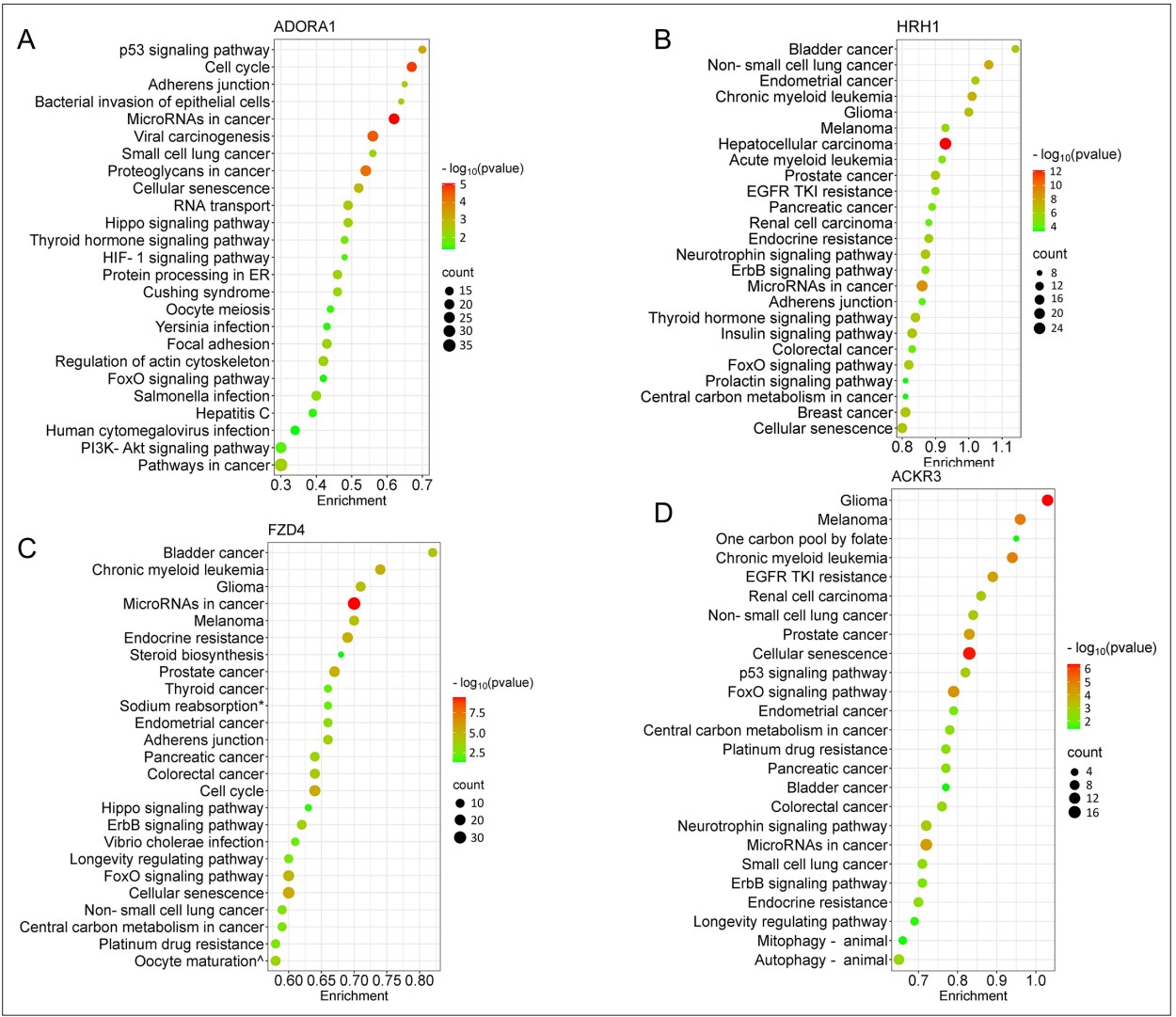

**Figure 4.** Pathway analysis of the differentially expressed extracellular vesicle (EV) microRNA (miRNA) after G protein-coupled receptor (GPCR) activation. The bubble plots show the top 25 significantly enriched Kyoto Encyclopedia of Genes and Genomes (KEGG) pathways for the miRNAs (≥1.5 fold change) of individual GPCR. (**A**) ADORA1, (**B**) HRH1, (**C**) FZD4, (**D**) ACKR3. The dot size represents the number of enriched gene targets, and the color shows the p-value of the enrichment. For the enrichment analysis, cutoff criteria were p-value (FDR)<0.05 and gene count >2. KEGG terms: * Endocrine and other factor-regulated calcium reabsorption, ^ Progesterone-mediated oocyte maturation.

The online version of this article includes the following figure supplement(s) for figure 4:

**Figure supplement 1.** The top 10 hub genes identified in microRNA (miRNA) targets (≥1.5 fold change) PPI network.

**Figure supplement 2.** Pathway analysis of miR-502-3p and miR-137 targets.

*2020*), we did not find significant changes in either the quantity or mean diameter between vehicle control and agonist-treated samples. We postulate that our EV production results could be due to the cell type and culture conditions (*Bost et al., 2022*; *Palviainen et al., 2019*). In addition, serum vesicles are a major source of EV contamination from the cell culture media. To minimize serum-derived EVs, our cells were cultured with EV-depleted FBS, which may have a reduced ability to support cell growth (*Bost et al., 2022*). Therefore, different isolation methods and cell culture conditions could result in variations in EV concentrations.

Few studies have addressed GPCR activation and its stimulation of EV release in cultured cells (*Alonso et al., 2011*; *Conrad et al., 2020*; *Mazzeo et al., 2016*; *Verweij et al., 2018*), indicating that GPCR signaling can regulate EV biogenesis and release. For instance, activation of (only) muscarinic receptor type I induces multivesicular body formation and stimulates EV secretion in T lymphocytes (*Alonso et al., 2011*). Similarly, stimulating (only) HRH1 promotes the exocytosis of multivesicular

bodies in HeLa cells (*Verweij et al., 2018*). Conrad et al. demonstrated that activation of multiple endogenous GPCRs stimulates EV release from trophoblasts (*Conrad et al., 2020*). Moreover, numerous studies have shown that the activation of GPCRs regulates the expression of individual miRNAs. For example, activation of the estrogen receptor inhibits miR-148a levels in breast cancer cells and COX-2 induces miR-526b expression through the prostaglandin receptor, which promotes breast cancer progression (*Majumder et al., 2015*; *Tao et al., 2015*). However, the aforementioned studies did not investigate EV miRNA expression after GPCR stimulation. Ambrose et al. identified released miRNA patterns that are highly correlated with specific GPCR agonists in platelets (*Ambrose et al., 2018*). However, their miRNA analysis used EV-rich supernatant after GPCR stimulation, which included both non-EV and EV miRNAs. Here, we focused on EV miRNA profiles following GPCR stimulation. We found many of the EV miRNAs were similar across the four GPCR subgroups, although some unique miRNAs were associated with each receptor. We identified 10 miRNAs in ADORA1, 4 miRNAs in HRH1, 10 miRNAs in FZD4, and 6 miRNAs in ACKR3 that were differentially expressed following agonist stimulation. We found that a greater number of miRNAs showed increased expression levels compared to those with decreased expression after receptor activation. The increase in miRNAs ranged from 6% to 30% more than decreased miRNAs, depending on the specific receptor group. For example, following ADORA1 activation, there were about 15% more increased than decreased miRNAs, while in the FZD4 group, there was a 6.3% higher number of increased miRNAs. Additionally, we observed a few miRNAs that exhibited more than a 1.5-fold change after each receptor activation. While endogenous miRNAs may be present in lower amounts in EVs, their functional significance can be considerable. Brown et al. demonstrated that a threshold concentration of miRNA is necessary to suppress target genes (*Brown et al., 2007*). This suggests that EV miRNAs may play a crucial role in miRNA expression in the recipient cells, helping them reach the required threshold to elicit specific effects.

Our study directly compared the stimulation of specific GPCR subclasses and the unique EV miRNA signatures that might dictate downstream $G\alpha_i$, $G\alpha_q$, $G\alpha_{12/13}$, and β-arrestin protein cascades. G proteins transduce the extracellular signals to the appropriate downstream effectors. Multiple GPCRs can couple to the same type of G protein. One GPCR also can interact with different G proteins to trigger multiple pathways. Previous reports showed that overexpression of Gα12 can alter multiple miRNA levels (*Yang et al., 2015*), suggesting that G proteins may play a role in controlling miRNA expression after GPCR stimulation. Despite our result suggesting a unique miRNA signature for each receptor, it is not clear that a miRNA signature was associated with specific types of G proteins due to the limited sample size (four receptors). Future studies are necessary to resolve whether EV miRNAs are regulated by specific G proteins after GPCR stimulation.

The mechanisms by which miRNAs are sorted into EVs are not fully understood. However, several RNA-binding proteins and membrane-associated proteins have been identified that could influence this sorting process (*Martins-Marques et al., 2022*; *Villarroya-Beltri et al., 2013*; *Yoon et al., 2015*). Argonaute 2 (Ago2), a core component of the RNA-induced silencing complexes, binds to miRNAs and facilitates their sorting. Ago2 can be regulated by the various cellular signaling pathways (*Horman et al., 2013*; *Jackson et al., 2022*; *McKenzie et al., 2016*). For instance, McKenzie et al. demonstrated that KRAS-dependent activation of MEK-ERK can phosphorylate the Ago2 protein, thereby facilitating the sorting of specific miRNAs into EVs (*McKenzie et al., 2016*). In differentiated PC12 cells, the activation of Gαq leads to the formation of Ago2-associated granules, which selectively sequester unique transcripts (*Jackson et al., 2022*). Another important molecular mediator is Connexin 43, which directly binds to specific miRNAs and sorts them into EVs (*Martins-Marques et al., 2022*). ACKR3, one of the GPCRs we studied here, when activated by an agonist, inhibits the function of Connexin 43 (*Fumagalli et al., 2020*). Furthermore, ADORA1 can directly interact with the membrane protein Caveolin 1, which has an established role in EV cargo selection (*Ginés et al., 2001*; *Lee et al., 2019*). Currently, there is no consistent mechanism for miRNAs sorting into EVs across different cell types. However, GPCRs and downstream signaling can directly or indirectly interact with those proteins involved in the sorting process, which could employ distinct mechanisms to influence the EV miRNA content. Future studies exploring the role of GPCRs, G proteins, and GPCR signaling in the expression, localization, activity, and post-translational modification of molecular mediators, such as Ago2 and Connexin 43, could provide valuable insights into how GPCRs regulate specific miRNAs within EVs and potentially lead to therapeutic applications.

When we compared the predicted functional effects of EV miRNAs across the four GPCRs, we found that numerous enriched pathways were associated with human cancers. For instance, the targets of HRH1 miRNA were enriched in colorectal cancer, breast cancer, and melanoma, which were consistent with previous findings (*Li et al., 2022*; *Park et al., 2023*; *Shi et al., 2019*). This could be because many of the predicted miRNA targets that are involved in a wide range of cellular activities, such as cell proliferation, differentiation, migration, and apoptosis, that contribute to cancer initiation, progression, and metastasis (*Evan and Vousden, 2001*; *Matthews et al., 2022*; *Pfeffer and Singh, 2018*). Additionally, the top hub genes identified in the PPI network of FZD4 miRNA targets, including MYC, EGFR, MDM2, and CDK1 (*Figure 4—figure supplement 1C*), were also involved in many aspects of cancer progression (*Gabay et al., 2014*; *Oliner et al., 2016*; *Sigismund et al., 2018*; *Wang et al., 2023*). This finding may contribute to understanding the mechanism of GPCR activities in cancer progression and to the development of new pharmacotherapies.

Moreover, our data revealed functional differences among the four GPCRs. When we compared the differentially expressed miRNAs and their predicted pathways across four GPCRs, we found ADORA1 had a unique set of miRNAs that did not overlap with the other three GPCRs. Three significantly altered ADORA1 miRNAs, miR-135a-5p, miR-31-5p, and miR-454-3p, are associated with cardioprotection (*Chen et al., 2022*; *Liu et al., 2024*; *Wang et al., 2021*; *Zhao et al., 2024*), supporting the established role of ADORA1 in the protection of cardiomyocytes during ischemia (*Crawford et al., 2005*; *Reichelt et al., 2005*). We observed that several insulin-resistant pathways were enriched among the targets of HRH1 miRNAs. The two significantly elevated HRH1 miRNAs, miR-502-3p and miR-423-5p, affect glucose metabolism (*Liu et al., 2023*; *Montastier et al., 2019*; *Yang et al., 2017*). Notably, miR-423-5p has been identified as a potential biomarker of severe obesity (*Ortega et al., 2013*). Consistent with previous studies (*Shi et al., 2019*; *Wang et al., 2010*), our findings indicate that HRH1 may play a role in regulating insulin sensitivity. MiR-137 and miR-518e-3p were solely expressed in EVs from cells treated with FZD4 agonist, and not after activation of other receptors, suggesting they may serve as unique signatures of FZD4 activation. Interestingly, genome-wide association and proteomic studies have implicated the *MIR137* host gene, and the mRNA targets of miR-137 are risk genes for schizophrenia (*Palumbo et al., 2023*; *Ripke et al., 2014*). Although the FZD activation mechanism and FZD-G protein interaction are still not fully understood (*Arthofer et al., 2016*; *Strakova et al., 2017*; *Zhang et al., 2024*), this study could lead to the identification of new miRNA targets for GPCR activation and post-transcriptional mRNA gene targets as a downstream effector for rational drug design. The predicted pathways for ACKR3 miRNA indicated that 9 of the top 25 significantly enriched pathways were associated with aging and neurodegenerative diseases. Cell senescence, which is a permanent cell growth arrest and linked to aging and several neurodegenerative disorders (*Kritsilis et al., 2018*; *Papadopoulos et al., 2020*; *Suelves et al., 2023*), was observed with the highest gene count among those pathways. Additionally, miR-135b, the only significantly increased ACKR3 miRNA, promotes hippocampal cell proliferation and enhances learning and memory by affecting the rate-limiting enzyme for β-amyloid production. miR-135b is significantly reduced in the peripheral blood of Alzheimer's disease patients (*Zhang et al., 2016*). GPCRs play a significant role in various diseases through different mechanisms of action. Our study suggests that the alteration of specific miRNA expression following GPCR activation by pharmacotherapies may introduce a new mechanism through which GPCRs regulate extracellular events by modulating the expression of miRNAs in EVs.

This study highlights the novel mechanism of EV miRNA content regulated by GPCRs. However, there are limitations in the current study. The EVs used for miRNA analysis are a heterogeneous population of various but similar sizes and may originate from different biogenesis pathways and exhibit distinct biological functions. GPCR agonists (or vehicles) may stimulate production of different subclasses of the EVs, such as exomeres, exosomes, or even microvesicles, and affect their content. Further experiments with optimized isolation methods are needed to investigate changes in EV subpopulations and miRNAs regulated by GPCRs. U2OS cells expressing endogenous GPCRs were used as a model to explore EV miRNA profiles. Thus, the differentially expressed miRNAs detected in the current study could be cell-type specific. EV content and miRNA expression could vary by cell type and culture conditions. Further research should investigate EV miRNA expression changes after receptor stimulation across different cell lines, particularly those related to diseases. For example, MDA-MB-231 is a breast cancer cell line that expresses several GPCRs, including adenosine, histamine,

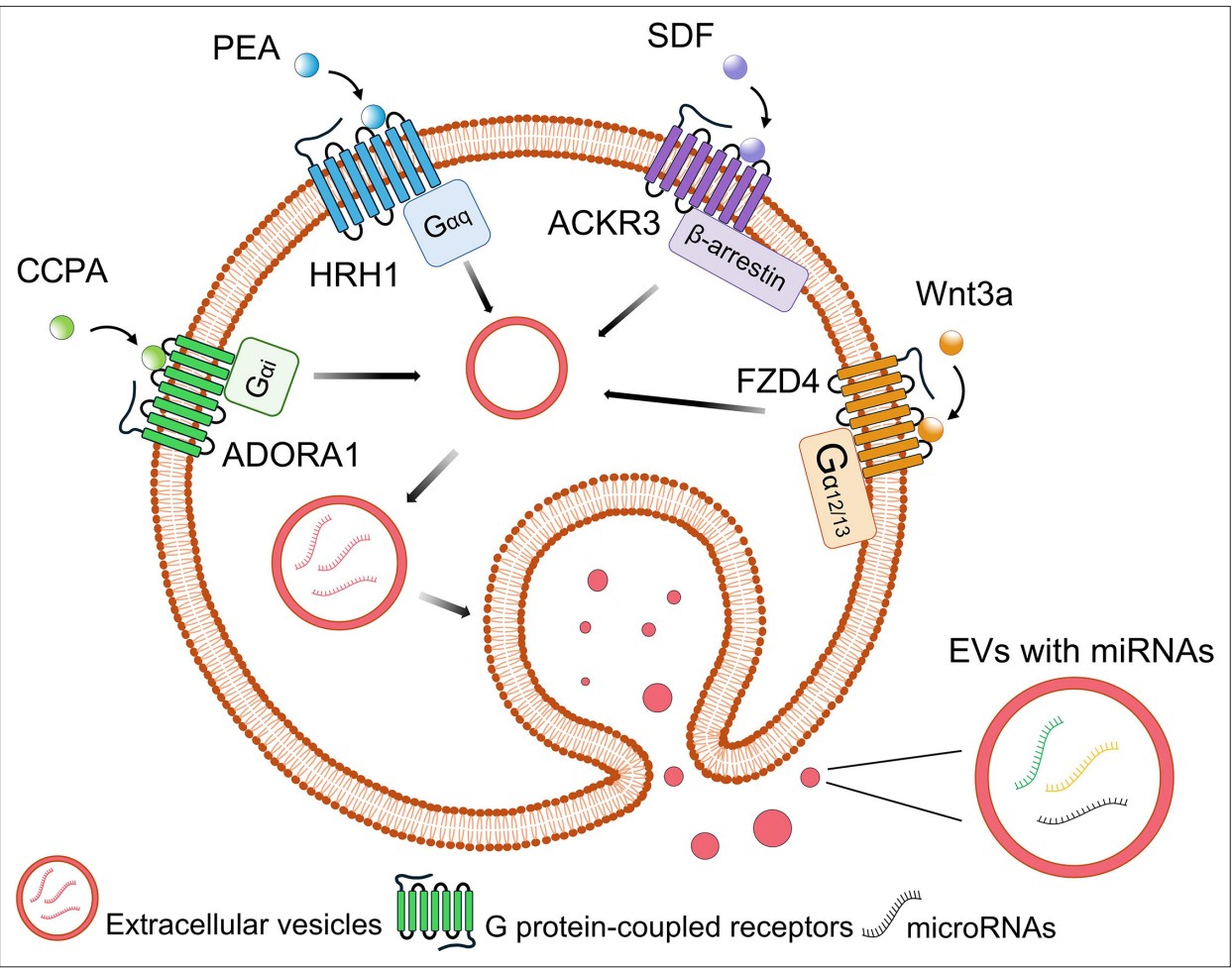

**Figure 5.** Stimulation of G protein-coupled receptors (GPCRs) activates extracellular vesicle (EV) microRNA signatures.

adhesion, and chemokine receptors (*Panjehpour et al., 2005*; *Park et al., 2023*; *Sarmoko et al., 2025*). Utilizing this cell line could provide valuable insights into the EV miRNA pattern associated with breast cancer in response to GPCR stimulation.

In this exploratory study, we used bioinformatics to predict the target genes and enriched pathways of the differentially expressed miRNAs following GPCR activation. While computational analyses can effectively identify potential miRNA targets, future research needs to focus on validating these candidates. For instance, the pathways enriched for miR-137, the most decreased miRNA following frizzled receptor activation, were found to be involved in GABAergic synapses, dopaminergic synapses, long-term potentiation, and autophagy (*Figure 4—figure supplement 2B*). These enriched pathways are essential for nerve activity and synaptic plasticity, which is consistent with the crucial role of FZD4 in brain development (*Zhang et al., 2017*; *Zhang et al., 2023*). Notably, an independent study from our laboratory confirmed that the delivery of miR-137 regulates synaptic proteins in both neuronal cell cultures and the mouse frontal cortex (*Palumbo et al., 2023*), which demonstrates the biological effect of miR-137 and its gene targets, aligning with our pathway prediction. Our study might illustrate a potential pathway for miRNAs and their targets in response to various stimuli. Future directions should aim to validate the top differentially expressed miRNAs identified after each GPCR activation and assess their gene targets in cellular contexts. Additionally, using proteomics to evaluate the molecular changes in an in vivo model following miRNA delivery will help enhance our understanding of the underlying mechanisms and explore the functional implications for clinical relevance.

In summary, we characterized EV miRNA profiles from U2OS cells after stimulation of specific GPCR subclasses with selective agonists (*Figure 5*). We identified unique EV miRNA signatures resulting from GPCR subtype activation. The functional analyses revealed that the predicted gene targets of

the EV miRNAs were enriched in distinct metabolic pathways, including physiological and pathological states of cell function. This study offers a novel biological mechanism, EV miRNA content regulation by stimulated GPCRs, and provides insight for future studies of drug-receptor interactions, including unwanted and unintended effects.

# Materials and methods

## Key resources table

| Reagent type (species) or resource | Designation | Source or reference | Identifiers | Additional information |
|---|---|---|---|---|
| Cell line (*Homo sapiens*) | U2OS (epithelial, osteosarcoma) | ATCC | HTB-96 RRID:CVCL_0042 | |
| Antibody | Rabbit polyclonal anti-Flotillin1 | Abcam | Ab41927 RRID:AB_941621 | 1:1000 |
| Antibody | Rabbit monoclonal anti-Syntenin | Abcam | Ab133267 RRID:AB_11160262 | 1:1000 |
| Antibody | Mouse monoclonal anti-CD63 | Santa Cruz Biotechnology | sc-5275 RRID:AB_627877 | 1:1000 |
| Antibody | Mouse monoclonal anti-CD81 | Santa Cruz Biotechnology | sc-166029 RRID:AB_2275892 | 1:500 |
| Antibody | Rabbit polyclonal anti-CD9 | Abcam | ab223052 RRID:AB_2922392 | 1:1000 |
| Antibody | Rabbit polyclonal anti-calnexin | Abcam | ab22595 RRID:AB_2069006 | 1:1000 |
| Antibody | Rabbit monoclonal anti-phospho-ERK1/2 (pERK1/2) | Millipore | 05-797R RRID:AB_1587016 | 1:1000 |
| Antibody | Rabbit polyclonal anti-p44/42 MAPK (total ERK1/2) | Cell Signaling Technology | 9102 RRID:AB_330744 | 1:1000 |
| Antibody | Mouse monoclonal anti-β-actin | Santa Cruz Biotechnology | sc-69879 RRID:AB_1119529 | 1:1000 |
| Sequence-based reagent | Megaplex RT primer human pool set v3.0 | Thermo Fisher Scientific | 4444750 | |
| Sequence-based reagent | Megaplex PreAmp primers human pool set v3.0 | Thermo Fisher Scientific | 4444748 | |
| Sequence-based reagent | TaqMan Array Human MicroRNA A+B Card Set v3.0 | Thermo Fisher Scientific | 4444913 | |
| Peptide, recombinant protein | Human recombinant SDF-1α | MilliporeSigma | GF344 | |
| Peptide, recombinant protein | Human recombinant Wnt3a | MilliporeSigma | H17001 | |
| Commercial assay or kit | cAMP EIA kit | Cayman Chemical | 581001 RRID:AB_3095671 | |
| Commercial assay or kit | ALP assay kit | Abcam | ab83369 | |
| Commercial assay or kit | Cisbio IP-1 Elisa kit | Revvity | 72IP1PEA RRID:AB_2904131 | |
| Commercial assay or kit | MagMax mirVana total RNA isolation kit | Applied Biosystems, Thermo Fisher Scientific | A27828 | |
| Commercial assay or kit | Qubit miRNA assay kit | Thermo Fisher Scientific | Q32880 | |
| Chemical compound, drug | Di8-ANEPPS | Thermo Fisher Scientific | D3167 | |
| Chemical compound, drug | Fluronic F-127 | Thermo Fisher Scientific | P3000MP | |
| Chemical compound, drug | 2-Chloro-$N^6$-cyclopentyladenosine (CCPA) | Tocris Bioscience | 1705 CAS:37739-05-2 | |
| Chemical compound, drug | 8-Cyclopentyl-1,3-dipropylxanthine (DPCPX) | Tocris Bioscience | 0439 CAS:102146-07-6 | |
| Chemical compound, drug | 2-Pyridylethylamine dihydrochloride | Tocris Bioscience | 2478 CAS:3343-39-3 | |

*Continued on next page*

*Continued*

| Reagent type (species) or resource | Designation | Source or reference | Identifiers | Additional information |
|---|---|---|---|---|
| Chemical compound, drug | Cetirizine dihydrochloride | Tocris Bioscience | 2577<br>CAS:83881-52-1 | |
| Software, algorithm | GraphPad Prism software version 10 | GraphPad Software Inc | RRID:SCR_002798 | |
| Software, algorithm | String (v12.0) | http://string.embl.de/ | RRID:SCR_005223 | |
| Software, algorithm | Gene Ontology | http://www.geneontology.org/ | RRID:SCR_002811 | |
| Software, algorithm | SR plot web service | https://www.bioinformatics.com.cn/srplot | RRID:SCR_025904 | |
| Software, algorithm | miRNet online web service v2.0 | http://www.mirnet.ca | RRID:SCR_024567 | |
| Software, algorithm | CytoScape software (v3.10.1) | https://cytoscape.org | RRID:SCR_003032 | |

## Drugs and reagents

CCPA, the selective adenosine $A_1$ receptor agonist, and adenosine $A_1$ receptor antagonist, DPCPX, were obtained from Tocris Bioscience (Minneapolis, MN, USA). PEA, the agonist of the histamine $H_1$ receptor, and cetirizine dihydrochloride, the antagonist of $H_1$ receptor, were obtained from Tocris Bioscience (Minneapolis, MN, USA). CCPA and DPCPX were diluted in DMSO, then subsequently diluted in the cell culture medium with a final DMSO concentration of 0.01%. Both PEA and cetirizine dihydrochloride were diluted in water. Human recombinant Wnt3a protein and SDF-1α were obtained from Sigma (Burlington, MA, USA). Wnt3a and SDF-1α were reconstituted to 100 µg/ml in sterile 1× PBS containing 0.1% human serum albumin (HSA) at 100 µg/ml. Wnt3a and SDF-1α were subsequently diluted in the cell culture medium with a final HSA concentration of 0.0001%. Gibco exosome-depleted FBS was purchased from Thermo Fisher (A2720801, Thermo Fisher Scientific, Waltham, MA, USA). Then, the FBS was further charcoal-stripped to remove the potential GPCR agonists in the serum.

## Cell culture

U2OS cells purchased from ATCC (Cat. No. HTB-96) were authenticated using STR profiling and tested negative for mycoplasma contamination. Cells were grown in McCoy 5A medium (16600082, Gibco, Thermo Fisher Scientific) supplemented with 10% fetal clone serum (FCS) and 1% penicillin and streptomycin and maintained in a humidified incubator with 10% $CO_2$ according to ATCC instructions. For EV isolation, cells were plated at $6 \times 10^6$ in a 15 cm culture dish 2 days before the experiment. On the day of the experiment, cells were rinsed with 1× PBS, then treated with agonists or vehicle control in the medium containing 10% EV-depleted and charcoal-stripped FBS. The treatments were left on for 24 hr before medium collection for EV isolation. Four 15 cm plates with a total medium volume of 55 ml of each condition were used for EV isolation.

## cAMP accumulation assay

U2OS cells were plated at $4 \times 10^5$/well in the 24-well plate 2 days before the assay. One day before the assay, cells were switched to the culture medium containing 10% charcoal-stripped FCS and incubated overnight. The assay procedure and buffer are as previously described (*Watts et al., 1998*). CCPA ($10^{-9}$ to $10^{-4}$) was added to the cells, and the plates were incubated for 20 min at 37°C in a humidified incubator with 5% $CO_2$. For the experiment with the antagonist, DPCPX (5 µM) was added to the cells and preincubated for 10 min before CCPA addition. cAMP accumulation was measured using a cAMP EIA kit from Cayman Chemical (581001, Ann Arbor, MI, USA), according to the manufacturer's instructions. All the experiments were conducted with three biological repeats, each performed with duplicated determinants.

## Inositol monophosphate formation

U2OS cells were plated at $4 \times 10^5$/well in the 24-well plate 1 day before the assay in the culture medium containing 10% charcoal-stripped FCS. The next day, cells were starved with plain McCoy 5A medium for 1 hr before the assay. Cells were preincubated with stimulation buffer for 10 min before agonist addition. Pyridylethylamine ($10^{-8}$ to $10^{-3}$) was added to the cells, and the plates were

incubated for 60 min at 37°C in a humidified incubator with 5% $CO_2$. For the experiment with the antagonist, cetirizine (1 µM) was preincubated for 10 min before agonist addition. The accumulation of inositol monophosphate was measured using the Cisbio IP-1 Elisa kit (72IP1PEA, Revvity, Waltham, MA, USA) as described previously (*Eshleman et al., 2013*). All the experiments were conducted with three independent repeats, each performed with duplicate determinants.

## ALP assay

U2OS cells were seeded at $4 \times 10^5$/well in the 12-well plate 1 day before the agonist treatment. Cells were treated with vehicle control (HSA) or various doses of Wnt3a diluted in cell culture medium and incubated for 24 hr. On the day of the assay, cells were rinsed with ice-cold 1× PBS and kept on the wet ice all the time. Cells were scraped and collected with 100 µl assay buffer/well, then transferred to microcentrifuge tubes. The cell suspensions were homogenized for 15–20 s with the micro pestle homogenizer. The lysates were centrifuged 13,000×g for 20 min at 4°C to remove insoluble material. ALP activity was measured using the ALP assay kit (ab83369, Abcam, Waltham, MA, USA). Briefly, 80 µl of cell lysate aliquot/well was mixed with kit reagent, then the plate was incubated at 25°C for 60 min. 10 µl of cell lysate was used to measure total protein concentration, which is used to normalize the enzyme activity. All the experiments were conducted with three independent repeats, each performed with duplicate determinants.

## EV isolation by SEC

To determine which fractions are EV-enriched, 30 ml of the medium control (plain medium with 10% EV-depleted FBS) and 30 ml of the conditioned medium from two 15 cm plates of U2OS cells incubated for 24 hr were collected and centrifuged at 300×g at 4°C for 15 min to remove cells and cell debris. The EV isolation methods were adapted from *Benedikter et al., 2017*. The EV characterization followed the Minimal Information for Studies of Extracellular Vesicles (MISEV) 2023 guidelines (*Welsh et al., 2024*). The supernatants were collected and centrifuged at 4000×g at 4°C for 15 min to further remove cell debris. After the second spin, the supernatants were passed through a 0.22 µM Millipore Steriflip filter (Burlington, MA, USA). Then, media were loaded on a 2% Tween precoated Amicon Ultra-15 centrifugal filter unit with Ultracel-10 membrane (MWCO = 10 kDa, Millipore, Burlington, MA, USA). The media were concentrated to ~550 µl by repeated centrifugation (4000×g) at 4°C. The concentrated media were fractionated by using SEC (Izon qEV Original/35 nM, IZON Science, Medford, MA, USA) according to the manufacturer's instructions. Briefly, 500 µl of concentrated media were loaded to the qEV column. 12 fractions of 0.5 ml each eluted with 1× PBS (0.1 uM filtered) were collected. Individual fractions were concentrated with Microcon-30 kDa Centrifugal Filters (Millipore, Burlington, MA, USA) to a final volume of 70 µl and stored at –80°C for immunoblot.

For further miRNA assessment, 60 ml of conditioned media pooled from four 15 cm plates per condition were harvested after 24 hr incubation of U2OS cells with agonists or vehicle control. For each GPCR, the experiments included 6 replicates of the agonist treatment group, 6 replicates of vehicle control, and 6–8 replicates of media control. The EV-enriched fractions 7, 8, and 9 (with a total volume of 1.5 ml) were pooled for TEM, NTA, and total RNA isolation. Pooled fractions were aliquoted and stored at –80°C until processed. Fractions used for TEM were stored at 4°C.

## Immunoblotting

To evaluate the EV-enriched SEC fractions, 500 µl of void (pooled fractions 1–6) and individual fractions 7–12 were concentrated with Microcon-30 kDa Centrifugal Filters (Millipore, Burlington, MA, USA) to a final volume of 70 µl. Due to the undetectable protein amount in the concentrated fractions, an equal volume (30 µl) for each fraction was subjected to SDS-PAGE through 12% polyacrylamide gels and transferred to polyvinylidene fluoride membranes (Bio-Rad, Hercules CA, USA). Membranes were blocked with 5% non-fat dry milk in TBST at room temperature for 30 min, and then subsequently incubated at 4°C overnight with rabbit anti-flotillin (1:1000, ab41927, Abcam, Waltham, MA, USA), rabbit anti-syntenin (1:1000, ab133267, Abcam, Waltham, MA, USA), mouse anti-CD63 (1: 1000, sc-5275, Santa Cruz Biotechnology), mouse anti-CD81 (1:500, sc-166029, Santa Cruz Biotechnology), rabbit anti-CD9 (1: 1000, ab223052, Abcam, Waltham, MA, USA), rabbit anti-calnexin (1:1000, ab22595, Abcam, Waltham, MA, USA). The immunoblots were rinsed and incubated with HRP-conjugated

secondary antibodies. Immune complexes were visualized by chemiluminescence and analyzed using Bio-Rad ChemiDoc Imager. Bands were analyzed by densitometry using Bio-Rad Image Lab software.

For the ACKR3 functional assay, cell lysates with a total of 15 µg protein of each condition were loaded into the gel. The immunoblots were incubated overnight with rabbit anti-pERK1/2 (1:1000, 05-797R, MilliporeSigma, Burlington BA, USA), rabbit anti-tERK1/2 (1:1000, 9102, Cell Signaling Technology, Danvers MA, USA), and mouse anti-β-actin (1:1000, sc-69879, Santa Cruz Biotechnology, Dallas, TX, USA). β-Actin was used as an internal control.

## TEM of EVs

15 µl of EV pooled fractions 7–9 were mixed with 5 µl of 16% paraformaldehyde and kept on the wet ice until processed. 5 µl of EV preparations were deposited and incubated for 3 min onto glow-discharged copper grids (Ted Pella 01822-F). After this incubation period, the grids were rinsed for 15 s in distilled water, wicked on Whatman filter paper #1, and then stained for 3 min in freshly prepared 1% (wt/vol) aqueous uranyl acetate. Grids were then wicked with filter paper and air-dried. Samples containing EVs were imaged at 120 kV on a FEI Tecnai Spirit TEM system. Images were acquired on an Advanced Microscopy Techniques (AMT) Nanosprint12 cMOS 12-megapixel camera system as tiff files.

## Fluorescence nanoparticle tracking analysis

SEC EVs (pooled void volume fractions 1–6 and pooled EV enriched fractions 7–9) from controlled media and media from U2OS cells were analyzed for particle concentration and size by ZetaView nanoparticle tracking analyzer (Particle Metrix, Ammersee, Germany) as described by the manufacturer's protocol. Briefly, EV samples were labeled with fluorescent membrane dye Di8-ANEPPS (D3167, Thermo Fisher Scientific, Waltham, MA, USA), staining solution (12 µM) supplemented with Fluronic F-127 (P3000MP Thermo Fisher Scientific, Waltham, MA, USA) at room temperature for 15 min. The samples were diluted at 1:200 in 0.1 µM filtered 0.1× PBS in a final volume of 1 ml for injection. Before the measurement, the instrument was calibrated with 100 nm polystyrene beads (Particle Metrix, Cat. No. 110-0020). The EV samples were recorded for 90 s at controlled room temperature. All measurements were done in triplicates. ZetaView software was used to quantify the concentration (particles/ml) and size distribution (nm) of the particles.

## EV RNA isolation and miRNA array

Pooled EV-enriched fractions (7, 8, and 9, total of 1.3 ml) were concentrated with Microcon-30 kDa Centrifugal Filters (Millipore, Burlington, MA, USA) to a final volume of 250 µl and used for EV RNA isolation. The total RNAs were isolated using MagMax mirVana total RNA isolation kit (A27828, Applied Biosystems, Thermo Fisher Scientific) according to the manufacturer's instructions. The miRNA concentration of each sample was measured with the Qubit miRNA assay kit (Q32880, Thermo Fisher Scientific). The isolated RNA was stored at –80°C until processing. The reverse transcription and pre-amplification reactions were run as previously described (*Sandau et al., 2020*). Briefly, RNA (with 8–10 ng of miRNA) was reverse-transcribed using Megaplex RT primer human pool set v3.0 (4444750, Thermo Fisher Scientific) and MultiScribe Reverse Transcriptase (4311235, Thermo Fisher Scientific) according to the user guide. To increase the sensitivity of qPCR analysis for miRNA targets, cDNA was pre-amplified (14 cycles) with Megaplex PreAmp primers human pool set v3.0 (4444748, Thermo Fisher Scientific) and TaqMan PreAmp Master mix (4391128, Thermo Fisher Scientific). Before the qPCR, pre-amplified cDNA was diluted 1:1 in nuclease-free water, then mixed with TaqMan Universal Master Mix (444047, Thermo Fisher Scientific). The samples were loaded to TaqMan Array Human MicroRNA A+B Card Set v3.0 (4444913, ThermoFisher Scientific), which contains a total of 754 miRNA assays. The PCR amplification and data acquisition were carried out using the QuantStudio 12K Flex Real-Time PCR system (Thermo Fisher Scientific).

## miRNA array and analysis

MiRNA expression was analyzed using relative quantification methods as previously described (*Sandau et al., 2020*). The Cq value for each well, along with the amplification score (AmpScore) and Cq confidence (CqConf), was reported by ExpressionSuit Software v1.3 (Thermo Fisher). Amplifications were filtered prior to the data analysis, as follows: PCR products with Cq >34, CqConf <0.8, or AmpScore

<1 were excluded and treated as missing values. Only expressed miRNAs meeting these criteria in more than 80% of both vehicle control and treatment group samples were accepted for further analysis. To maximize accuracy and minimize amplification variation between the A and B cards, a mean expression value calculated from qualified miRNAs pooled from both the A and B cards was used to normalize the PCR data. miRNAs chosen for calculating mean expression for endogenous background normalization had to meet the following criteria: (1) the miRNA was not expressed or was observed in less than 50% of the untreated media, (2) was consistently fully observed in all vehicle control and treated samples, and (3) for miRNAs observed at rates of less than 50% in untreated media, the Cq value of the media control was required to be >3.3 cycles different from the vehicle control. The mean expression Cq value of all the qualified miRNAs was calculated and used for endogenous background normalization. The $\Delta$Cq value for a miRNA was calculated as: $\Delta$Cq = Cq value of the miRNA − mean normalizer Cq value in the same group. The $\Delta\Delta$Cq value for a miRNA was calculated as: $\Delta\Delta$Cq = $\Delta$Cq for treated samples − $\Delta$Cq for vehicle control samples. The fold change (relative quantification value) for each miRNA was calculated as: $2^{-\Delta\Delta Cq}$. A relative quantification value >1 indicates increased miRNA expression in the treated samples compared to vehicle control, and a relative quantification value <1 indicates decreased miRNA expression in the treated samples.

## Data analysis and statistical analysis

Dose-response curves for cAMP and IP1 accumulation were analyzed by nonlinear regression. Significant differences were determined by two-way analysis of variance (ANOVA) of receptors by agonist or antagonist concentration followed by Sidak's multiple comparison test. For the ALP accumulation and ERK1/2 activation, the significant differences were assessed by one-way ANOVA followed by Tukey's multiple comparison test. All functional assay data were shown as mean ± SD. The statistical significance in EV concentrations was determined using Student's t-test. Data were expressed as mean ± SEM. Data were analyzed with GraphPad Prism software v10 (GraphPad Software Inc, San Diego, CA, USA). Statistical tests were applied based on three independent replicates. p-Values<0.05 were considered significant.

To statistically evaluate the qPCR miRNA expression data, we elected to use strongly unbiased robust procedures, including matched-pairs estimators of the effect sizes and rank-based testing of associations in the interest of minimizing confounding of associations by technical causes, and then to further interrogate those results with extensive sensitivity analyses designed to probe the volatility of the conclusions depending on our choice of analytical methods. For example, as an alternative normalization procedure to relax the assumption (needed for validity of per-run normalization) of no differential measurement drift and to mitigate the run-to-run volatility of normalizer mean estimation, we considered aligning median normalizer scores by treatment group rather than explicitly adjusting qPCR runs for each sample, but observed that residual technical variance (in the aligned normalizer scores) between matched pairs was actually increased due to the complications of missing data created by quality filtering, the technical run-batching of the assay into A and B cards, varying qPCR performance of even the endogenous normalizer miRNAs across runs, and other technical factors beyond our control.

To estimate the treatment effects on miRNA and calculate robust p-values for testing of those effects, we considered six different combinations of treatment effect estimator and p-value estimator in sensitivity analyses (*Figure 3—figure supplement 1*); each of the approaches offers an alternative estimate of the magnitude and significance of the $\Delta\Delta$Cq value, varying depending on how pairs and treatment groups and batches are aggregated or disaggregated. We consider the matched-pairs estimator (*Allison, 2009*) of treatment effect superior to other approaches because of its strong unbiasedness even in the presence of technical confounding, and the Skillings-Mack estimator (*Skillings and Mack, 1981*) of the p-value superior both to naïve permutation-based (*Good, 2006*) or rank-based tests (*Mann and Whitney, 1947*) and to Welch's robust t-test (*Welch, 1947*) because it explicitly accounts for imbalanced group sizes (in our case caused by missing data at the per-miRNA level).

Ranking of miRNAs within each receptor type was first performed by magnitude of treatment effect (in normalized $\Delta$Cq units), as estimated from the matched-pairs regression approach, and the ranking was then filtered by the Skillings-Mack p-value. Because of the first-in-practice exploratory nature of this study, we opted to emphasize discovery power over avoidance of false positives and thus chose a liberal cutoff of p<0.2 for selection of associated miRNAs for further investigation.

Pathway enrichment and functional annotation analyses were limited to the subset of miRNAs highlighted as the most affected for each receptor; the cutoff for this was absolute fold change $\geq 1.5$ ($|\Delta\Delta Cq|\geq 0.585$) for agonist vs. vehicle control. The $\log_2$ fold change, Skillings-Mack p-values, and z-transformed $\log_2$ fold change values (Z-score) were used to create volcano plots and hierarchical heatmaps. The hierarchical clustering in the heatmap is based on Euclidean distance. The visualizations were generated using GraphPad Prism v10 and SR plot web service (https://www.bioinformatics.com.cn/srplot).

### miRNA targets and pathway analysis

Prediction of miRNA targets was conducted using miRNet online web service v2.0 (http://www.mirnet.ca). The miRNA-target interactions were analyzed using the experimentally validated miRNA-target interaction databases: miRTarbase (v8.0) and Tarbase (v8.0). Human transcripts targeted by at least two miRNAs from each receptor were exported for further functional analysis. The target gene lists were submitted to String (v12.0) and Gene Ontology with default settings to identify pathways enriched with miRNA targets and construct PPI networks. KEGG pathway enrichment analyses and WikiPathway analyses were performed with String (v12.0) using the hypergeometric algorithm, and the statistically enriched terms were determined by Fisher's exact test with adjusted p-value (false discovery rate)<0.05 as the threshold. To measure the enrichment effect, the pathways were sorted by $\log_{10}$(observed/expected), which is the ratio between the number of proteins in our observed network that are annotated with a term and the number of the proteins that the software expects to be annotated with the same term in a random network of the similar size. Pathways exhibiting significant changes were further visualized with bubble plots using the SR plot web service (https://www.bioinformatics.com.cn/srplot). The PPI networks constructed by String were analyzed with CytoScape software (v3.10.1). The hub genes were identified and selected using CytoHubba, a plug-in of CytoScape, according to the node degree in the PPI network.

## Acknowledgements

We thank Dr. Alfred Lewy, Dr. Kim Neve, and Dr. Atheir Abbas for critically reading the manuscript. We also acknowledge the OHSU Gene Profiling Shared Resource for providing the technical support for the qPCR assays. Transmission electron microscopy was performed at the Multiscale Microscopy Core, a member of the OHSU University Shared Resource Cores (RRID:SCR_022652) under the guidance of Dr. Claudia S Lopez.

## Additional information

### Funding

| Funder | Grant reference number | Author |
|---|---|---|
| National Institute on Drug Abuse | ADA 12013 | Aaron J Janowsky |
| United States Food and Drug Administration | CDER-20-I-0546 | Aaron J Janowsky |
| Drug Enforcement Administration | D-15-0D-002 | Aaron J Janowsky |
| United States Department of Veterans Affairs | 1IK6BX005754 | Aaron J Janowsky |
| National Institute on Drug Abuse | T32AG055378-06 | Michelle C Palumbo |

The funders had no role in study design, data collection and interpretation, or the decision to submit the work for publication.

## Author contributions

Xiao Shi, Conceptualization, Data curation, Formal analysis, Investigation, Visualization, Methodology, Writing – original draft, Writing – review and editing; Michelle C Palumbo, Conceptualization, Investigation, Methodology, Writing – original draft, Writing – review and editing; Sheila Benware, Sheila Markwardt, Investigation, Methodology, Writing – review and editing; Jack Wiedrick, Formal analysis, Investigation, Visualization, Methodology, Writing – original draft, Writing – review and editing; Aaron J Janowsky, Conceptualization, Funding acquisition, Investigation, Methodology, Writing – original draft, Writing – review and editing

## Author ORCIDs

Xiao Shi ⬤ https://orcid.org/0000-0002-6944-9135

Reviewer #1 (Public review): https://doi.org/10.7554/eLife.107865.3.sa1
Reviewer #2 (Public review): https://doi.org/10.7554/eLife.107865.3.sa2
Author response https://doi.org/10.7554/eLife.107865.3.sa3

# Additional files

## Supplementary files

Supplementary file 1. The analysis of G protein-coupled receptor (GPCR) gene expression across various cell lines, as sourced from the Human Protein Atlas.

Supplementary file 2. microRNAs (miRNAs) detected in media control.

Supplementary file 3. Differentially expressed microRNAs (miRNAs) in the isolated extracellular vesicles (EVs) following G protein-coupled receptor (GPCR) activation. All differentially expressed miRNAs (meeting p<0.2) after stimulation were listed.

MDAR checklist

## Data availability

The microRNA array data is MIAME compliant and has been deposited to the Gene Expression Omnibus site: http://www.ncbi.nlm.nih.gov/geo/. GEO accession numbers: GSE270077, GSE270078, GSE270079, and GSE270080. All data needed to evaluate the conclusions in the paper are present in the paper or in *Supplementary files 1–3*.

The following datasets were generated:

| Author(s) | Year | Dataset title | Dataset URL | Database and Identifier |
|---|---|---|---|---|
| Shi X, Palumbo MC, Benware S, Wiedrick J | 2025 | Real-time quantitative PCR analysis of extracellular vesicle miRNA from human U2OS cells culture medium [ACKR3] | https://www.ncbi.nlm.nih.gov/geo/query/acc.cgi?acc=GSE270077 | NCBI Gene Expression Omnibus, GSE270077 |
| Shi X, Palumbo MC, Benware S, Wiedrick J | 2025 | Real-time quantitative PCR analysis of extracellular vesicle miRNA from human U2OS cells culture medium [AODRA1] | https://www.ncbi.nlm.nih.gov/geo/query/acc.cgi?acc=GSE270078 | NCBI Gene Expression Omnibus, GSE270078 |
| Shi X, Palumbo MC, Benware S, Wiedrick J | 2025 | Real-time quantitative PCR analysis of extracellular vesicle miRNA from human U2OS cells culture medium [FZD4] | https://www.ncbi.nlm.nih.gov/geo/query/acc.cgi?acc=GSE270079 | NCBI Gene Expression Omnibus, GSE270079 |
| Shi X, Palumbo MC, Benware S, Wiedrick J | 2025 | Real-time quantitative PCR analysis of extracellular vesicle miRNA from human U2OS cells culture medium [HRH1] | https://www.ncbi.nlm.nih.gov/geo/query/acc.cgi?acc=GSE270080 | NCBI Gene Expression Omnibus, GSE270080 |

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
