## [Editor Report · eLife Assessment]

This study presents **valuable** findings by demonstrating that specific GPCR subtypes induce distinct extracellular vesicle miRNA signatures, highlighting a potential novel mechanism for intercellular communication with implications for receptor pharmacology within the field. The evidence is **solid**, however, more experiments are needed to determine whether the distinct extracellular vesicle miRNA signatures result from GPCR-dependent miRNA expression or GPCR-dependent incorporation of miRNAs into extracellular vesicles.

---

## [Referee Report · Reviewer #1 (Public review)]

Summary:

GPCRs affect the EV-miRNA cargoes

Strengths:

Novel idea of GPCRs-mediated control of EV loading of miRNAs

Weaknesses:

Incomplete findings failed to connect and show evidence of any physiological parameters that are directly related to the observed changes. The mechanical detail is completely lacking.

Comments on revisions:

The revised version of the manuscript falls short of the required standard by lacking additional experiments. Some of the conditions for acceptability could have been met only through clarifying uncertainties via further experiments, which, unfortunately, have not been conducted.

---

## [Referee Report · Reviewer #2 (Public review)]

Summary:

This study examines how activating specific G protein-coupled receptors (GPCRs) affects the microRNA (miRNA) profiles within extracellular vesicles (EVs). The authors seek to identify whether different GPCRs produce unique EV miRNA signatures and what these signatures could indicate about downstream cellular processes and pathology processes.

Methods:

Used U2OS human osteosarcoma cells, which naturally express multiple GPCR types.

Stimulated four distinct GPCRs (ADORA1, HRH1, FZD4, ACKR3) using selective agonists.

Isolated EVs from culture media and characterized them via size exclusion chromatography, immunoblotting, and microscopy.

Employed qPCR-based miRNA profiling and bioinformatics analyses (e.g., KEGG, PPI networks) to interpret expression changes.

Key Findings:

No significant change in EV quantity or size following GPCR activation.

Each GPCR triggered a distinct EV miRNA expression profile.

miRNAs differentially expressed post-stimulation were linked to pathways involved in cancer, insulin resistance, neurodegenerative diseases, and other physiological/pathological processes.

miRNAs such as miR-550a-5p, miR-502-3p, miR-137, and miR-422a emerged as major regulators following specific receptor activation.

Conclusions:

The study offers evidence that GPCR activation can regulate intercellular communication through miRNAs encapsulated within extracellular vesicles (EVs). This finding paves the way for innovative drug-targeting strategies and enhances understanding of drug side effects that are mediated via GPCR-related EV signaling.

Strengths:

Innovative concept: The idea of linking GPCR signaling to EV miRNA content is novel and mechanistically important.

Robust methodology: The use of multiple validation methods (biochemical, biophysical, and statistical) lends credibility to the findings.

Relevance: GPCRs are major drug targets, and understanding off-target or systemic effects via EVs is highly valuable for pharmacology and medicine.

Weaknesses:

Sample Size & Scope: The analysis included only four GPCRs. Expanding to more receptor types or additional cell lines would enhance the study's applicability.

Exploratory Nature: This study is primarily descriptive and computational. It lacks functional validation, such as assessing phenotypic effects in recipient cells, which is acknowledged as a future step.

EV heterogeneity: The authors recognize that they did not distinguish EV subpopulations, potentially confounding the origin and function of miRNAs.

Comments on revisions:

All the comments have been taken into account. I wish the authors success in their future research.

---

## [Author Response]

The following is the authors’ response to the original reviews.

**Public Reviews:**

**Reviewer #1 (Public review):**
Summary:In this manuscript, the authors explore a novel concept: GPCR-mediated regulation of miRNA release via extracellular vesicles (EVs). They perform an EV miRNA cargo profiling approach to investigate how specific GPCR activations influence the selective secretion of particular miRNAs. Given that GPCRs are highly diverse and orchestrate multiple cellular pathways - either independently or collectively - to regulate gene expression and cellular functions under various conditions, it is logical to expect alterations in gene and miRNA expression within target cells.Strengths:The novel idea of GPCRs-mediated control of EV loading of miRNAs.Weaknesses:Incomplete findings failed to connect and show evidence of any physiological parameters that are directly related to the observed changes. The mechanical detail is lacking.

We appreciate the reviewer's acknowledgment of the novelty of this study. We agree with the reviewer that further mechanistic insights would strengthen the manuscript. The mechanisms by which miRNA is sorted into EVs remain poorly understood. Various factors, including RNAbinding protein, sequence motifs, and cellular location, can influence this sorting process(Garcia-Martin et al., 2022; Liu & Halushka, 2025; Villarroya-Beltri et al., 2013; Yoon et al., 2015). Ago2, a key component of the RNA-induced silencing complexes, binds to miRNA and facilitates miRNA sorting. Ago2 has been found in the EVs and can be regulated by the cellular signaling pathway. For instance, McKenzie et al. demonstrated that KRAS-dependent activation of MEK-ERK can phosphorylate Ago2 protein, thereby regulating the sorting of specific miRNAs into EVs(McKenzie et al., 2016). In the differentiated PC12 cells, Gαq activation leads to the formation of Ago2-associated granules, which selectively sequester unique transcripts(Jackson et al., 2022). Investigating GPCR, G protein, and GPCR signaling on Ago2 expression, location, and phosphorylation states could provide valuable insights into how GPCRs regulate specific miRNAs within EVs. We have expanded these potential mechanisms and future research in the discussion section (page 16-17).

The manuscript falls short of providing a comprehensive understanding. Identifying changes in cellular and EV-associated miRNAs without elucidating their physiological significance or underlying regulatory mechanisms limits the study's impact. Without demonstrating whether these miRNA alterations have functional consequences, the findings alone are insufficient. The findings may be suitable for more specialized journals.

Thank you for the feedback. We acknowledge that validating the target genes of the top candidate miRNAs is an important next step. In response to the reviewer's concerns, we have expanded the discussion of future research in the manuscript (page 19-20). Although this initial study is primarily descriptive, it establishes a novel conceptual link between GPCR signaling and EV-mediated communication.

Furthermore, a critical analysis of the relationship between cellular miRNA levels and EV miRNA cargo is essential. Specifically, comparing the intracellular and EV-associated miRNA pools could reveal whether specific miRNAs are preferentially exported, a behavior that should be inversely related to their cellular abundance if export serves a beneficial function by reducing intracellular levels. This comparison is vital to strengthen the biological relevance of the findings and support the proposed regulatory mechanisms by GPCRs.

We appreciate the valuable suggestions from the reviewer. EV miRNA and cell miRNAs may exhibit distinct profiles as miRNAs can be selectively sorted into or excluded from EVs(Pultar et al., 2024; Teng et al., 2017; Zubkova et al., 2021). Investigating the difference between cellular miRNA levels and EV miRNA cargo would provide insight into the mechanism of miRNA sorting and the functions of miRNAs in the recipient cells. The expression of the cellular miRNAs is a highly dynamic process. To accurately compare the miRNA expression levels, profiling of EV miRNA and cellular miRNA should be conducted simultaneously. However, as an exploratory study, we were unable to measure the cellular miRNAs without conducting the entire experiment again.

**Reviewer #2 (Public review):**
Summary:This study examines how activating specific G protein-coupled receptors (GPCRs) affects the microRNA (miRNA) profiles within extracellular vesicles (EVs). The authors seek to identify whether different GPCRs produce unique EV miRNA signatures and what these signatures could indicate about downstream cellular processes and pathological processes.Methods:(1) Used U2OS human osteosarcoma cells, which naturally express multiple GPCR types.(2) Stimulated four distinct GPCRs (ADORA1, HRH1, FZD4, ACKR3) using selective agonists.(3) Isolated EVs from culture media and characterized them via size exclusion chromatography, immunoblotting, and microscopy.(4) Employed qPCR-based miRNA profiling and bioinformatics analyses (e.g., KEGG, PPI networks) to interpret expression changes.Key Findings:(1) No significant change in EV quantity or size following GPCR activation.(2) Each GPCR triggered a distinct EV miRNA expression profile.(3) miRNAs differentially expressed post-stimulation were linked to pathways involved in cancer, insulin resistance, neurodegenerative diseases, and other physiological/pathological processes.(4) miRNAs such as miR-550a-5p, miR-502-3p, miR-137, and miR-422a emerged as major regulators following specific receptor activation.Conclusions:The study offers evidence that GPCR activation can regulate intercellular communication through miRNAs encapsulated within extracellular vesicles (EVs). This finding paves the way for innovative drug-targeting strategies and enhances understanding of drug side effects that are mediated via GPCR-related EV signaling.Strengths:(1) Innovative concept: The idea of linking GPCR signaling to EV miRNA content is novel and mechanistically important.(2) Robust methodology: The use of multiple validation methods (biochemical, biophysical, and statistical) lends credibility to the findings.(3) Relevance: GPCRs are major drug targets, and understanding off-target or systemic effects via EVs is highly valuable for pharmacology and medicine.Weaknesses:(1) Sample Size & Scope: The analysis included only four GPCRs. Expanding to more receptor types or additional cell lines would enhance the study's applicability.

We are encouraged that the reviewer recognized the novelty, methodological rigor, and significance of our work. We recognize the limitations of our current model system and emphasize the need to test additional GPCR families and cell lines in the future studies, as detailed in the discussion section (Page 19, second paragraph).

(2) Exploratory Nature: This study is primarily descriptive and computational. It lacks functional validation, such as assessing phenotypic effects in recipient cells, which is acknowledged as a future step.

We appreciate the feedback. We recognize the importance of validating the function of the top candidate miRNAs in the recipient cells, and this will be included in future studies (page 19-20).

(3) EV heterogeneity: The authors recognize that they did not distinguish EV subpopulations, potentially confounding the origin and function of miRNAs.

Thank you for the comment. EV isolation and purification are major challenges in EV research. Current isolation techniques are often ineffective at separating vesicles produced by different biogenetic pathways. Furthermore, the lack of specific markers to differentiate EV subtypes adds to this complexity. We recognize that the presence of various subpopulations can complicate the interpretation of EV cargos. In our study, we used a combined approach of ultrafiltration followed by size-exclusion chromatography to achieve a balance between EV purity and yield. We adhere to the MISEV (Minimal Information for Studies of Extracellular Vesicles 2023) guidelines by reporting detailed isolation methods, assessing both positive and negative protein markers, and characterizing EVs by electron microscopy to confirm vesicle structure, as well as nanoparticle tracking analysis to verify particle size distribution(Welsh et al., 2024). By following these guidelines, we can ensure the quality of our study and enhance the ability to compare our findings with other studies.

**Recommendations for the authors:**

**Reviewer #2 (Recommendations for the authors):**
Suggestions for Future Research:(1) Functionally validate top candidate miRNAs in recipient cells.

We acknowledge that validating the target genes of the top candidate miRNAs is a crucial next step. In response to the reviewer's concerns, we have included this in the discussion as future research in the manuscript (page 19-20).

(2) Investigate other GPCR families and repeat in primary or disease-relevant cell lines.

The inclusion of different GPCRs and cell lines is suggested as an area for further investigation in the discussion. (Page 19).

(3) Apply similar approaches in in vivo models or patient samples to assess clinical relevance.

In response to the reviewer's concerns, we have included this in the discussion as future research in the manuscript (page 19-20).

References

Garcia-Martin, R., Wang, G., Brandão, B. B., Zanotto, T. M., Shah, S., Kumar Patel, S., Schilling, B., & Kahn, C. R. (2022). MicroRNA sequence codes for small extracellular vesicle release and cellular retention. Nature, 601(7893), 446-451. https://doi.org/10.1038/s41586021-04234-3

Jackson, L., Rennie, M., Poussaint, A., & Scarlata, S. (2022). Activation of Gαq sequesters specific transcripts into Ago2 particles. Sci Rep, 12(1), 8758. https://doi.org/10.1038/s41598022-12737-w

Liu, X.-M., & Halushka, M. K. (2025). Beyond the Bubble: A Debate on microRNA Sorting Into Extracellular Vesicles. Laboratory Investigation, 105(2), 102206. https://doi.org/10.1016/j.labinv.2024.102206

McKenzie, A. J., Hoshino, D., Hong, N. H., Cha, D. J., Franklin, J. L., Coffey, R. J., Patton, J. G., & Weaver, A. M. (2016). KRAS-MEK Signaling Controls Ago2 Sorting into Exosomes. Cell Rep, 15(5), 978-987. https://doi.org/10.1016/j.celrep.2016.03.085

Pultar, M., Oesterreicher, J., Hartmann, J., Weigl, M., Diendorfer, A., Schimek, K., Schädl, B., Heuser, T., Brandstetter, M., Grillari, J., Sykacek, P., Hackl, M., & Holnthoner, W. (2024).Analysis of extracellular vesicle microRNA profiles reveals distinct blood and lymphatic endothelial cell origins. J Extracell Biol, 3(1), e134. https://doi.org/10.1002/jex2.134

Teng, Y., Ren, Y., Hu, X., Mu, J., Samykutty, A., Zhuang, X., Deng, Z., Kumar, A., Zhang, L., Merchant, M. L., Yan, J., Miller, D. M., & Zhang, H.-G. (2017). MVP-mediated exosomal sorting of miR-193a promotes colon cancer progression. Nature Communications, 8(1), 14448. https://doi.org/10.1038/ncomms14448

Villarroya-Beltri, C., Gutiérrez-Vázquez, C., Sánchez-Cabo, F., Pérez-Hernández, D., Vázquez, J., Martin-Cofreces, N., Martinez-Herrera, D. J., Pascual-Montano, A., Mittelbrunn, M., & Sánchez-Madrid, F. (2013). Sumoylated hnRNPA2B1 controls the sorting of miRNAs into exosomes through binding to specific motifs. Nat Commun, 4, 2980. https://doi.org/10.1038/ncomms3980

Welsh, J. A., Goberdhan, D. C. I., O'Driscoll, L., Buzas, E. I., Blenkiron, C., Bussolati, B., Cai, H., Di Vizio, D., Driedonks, T. A. P., Erdbrügger, U., Falcon-Perez, J. M., Fu, Q. L., Hill, A. F., Lenassi, M., Lim, S. K., Mahoney, M. G., Mohanty, S., Möller, A., Nieuwland, R., . . .Witwer, K. W. (2024). Minimal information for studies of extracellular vesicles (MISEV2023): From basic to advanced approaches. J Extracell Vesicles, 13(2), e12404. https://doi.org/10.1002/jev2.12404

Yoon, J. H., Jo, M. H., White, E. J., De, S., Hafner, M., Zucconi, B. E., Abdelmohsen, K., Martindale, J. L., Yang, X., Wood, W. H., 3rd, Shin, Y. M., Song, J. J., Tuschl, T., Becker, K. G., Wilson, G. M., Hohng, S., & Gorospe, M. (2015). AUF1 promotes let-7b loading on Argonaute 2. Genes Dev, 29(15), 1599-1604. https://doi.org/10.1101/gad.263749.115

Zubkova, E., Evtushenko, E., Beloglazova, I., Osmak, G., Koshkin, P., Moschenko, A., Menshikov, M., & Parfyonova, Y. (2021). Analysis of MicroRNA Profile Alterations in Extracellular Vesicles From Mesenchymal Stromal Cells Overexpressing Stem Cell Factor. Front Cell Dev Biol, 9, 754025. https://doi.org/10.3389/fcell.2021.754025